# The Role of Palliative Radiotherapy in the Treatment of Spinal Bone Metastases from Head and Neck Tumors—A Multicenter Analysis of a Rare Event

**DOI:** 10.3390/cancers12071950

**Published:** 2020-07-18

**Authors:** Tilman Bostel, Alexander Rühle, Tilmann Rackwitz, Arnulf Mayer, Tristan Klodt, Laura Oebel, Robert Förster, Ingmar Schlampp, Daniel Wollschläger, Harald Rief, Tanja Sprave, Jürgen Debus, Anca-Ligia Grosu, Heinz Schmidberger, Sati Akbaba, Nils Henrik Nicolay

**Affiliations:** 1Department of Radiation Oncology, University Medical Center Mainz, Langenbeckstrasse 1, 55131 Mainz, Germany; arnulf.mayer@unimedizin-mainz.de (A.M.); tristan.klodt@unimedizin-mainz.de (T.K.); laura.oebel@unimedizin-mainz.de (L.O.); heinz.schmidberger@unimedizin-mainz.de (H.S.); sati.akbaba@unimedizin-mainz.de (S.A.); 2Department of Radiation Oncology, University Hospital of Freiburg, Robert-Koch-Strasse 3, 79106 Freiburg, Germany; tanja.sprave@uniklinik-freiburg.de (T.S.); anca.grosu@uniklinik-freiburg.de (A.-L.G.); nils.nicolay@uniklinik-freiburg.de (N.H.N.); 3Department of Radiation Oncology, University Hospital of Heidelberg, Im Neuenheimer Feld 400, 69120 Heidelberg, Germany; tilmann.rackwitz@med.uni-heidelberg.de (T.R.); ingmar.schlampp@med.uni-heidelberg.de (I.S.); juergen.debus@med.uni-heidelberg.de (J.D.); 4Institute of Radiation Oncology, Cantonal Hospital Winterthur, University of Zurich, Brauerstrasse 15, 8401 Winterthur, Switzerland; robert.foerster@ksw.ch; 5Institute of Medical Biostatistics, Epidemiology and Informatics (IMBEI), University Medical Center Mainz, 55131 Mainz, Germany; wollschlaeger@uni-mainz.de; 6Radiotherapy Bonn-Rhein-Sieg, 53115 Bonn, Germany; harald.rief@gmx.at

**Keywords:** spinal bone metastases, instability, head and neck cancer, radiotherapy, SINS

## Abstract

This retrospective multi-center analysis aimed to assess the clinical response and stabilizing effects of palliative radiotherapy (RT) for spinal bone metastases (SBM) in head and neck cancer (HNC), and to establish potential predictive factors for stability and overall survival (OS). Patients included in this analysis were treated at the University Hospitals of Mainz, Freiburg, and Heidelberg between 2001 and 2019. Clinical information was taken from the medical records. The stability of affected vertebral bodies was assessed according to the validated spine instability neoplastic score (SINS) based on CT-imaging before RT, as well as 3 and 6 months after RT. OS was quantified as the time between the start of palliative RT and death from any cause or last follow-up. Potential predictive factors for stability and OS were analyzed using generalized estimating equations and Cox regression for time-varying covariates to take into account multiple observations per patient. The mean follow-up time of 66 included patients after the first palliative RT was 8.1 months (range 0.3–85.0 months). The majority of patients (70%; *n* = 46) had squamous cell carcinomas (SCC) originating from the pharynx, larynx and oral cavity, while most of the remaining patients (26%; *n* = 17) suffered from salivary glands tumors. A total of 95 target volumes including 178 SBM were evaluated that received a total of 81 irradiation series. In patients with more than one metastasis per irradiated region, only the most critical bone metastasis was analyzed according to the SINS system. Prior to RT, pain and neurologic deficits were present in 76% (*n* = 72) and 22% (*n* = 21) of irradiated lesions, respectively, and 68% of the irradiated lesions (*n* = 65) were assessed as unstable or potentially unstable prior to RT. SBM-related pain symptoms and neurologic deficits responded to RT in 63% and 47% of the treated lesions, respectively. Among patients still alive at 3 and 6 months after RT with potentially unstable or unstable SBM, a shift to a better stability class according to the SINS was observed in 20% and 33% of the irradiated SBM, respectively. Pathological fractures of SBM were frequently detected before the start of irradiation (43%; *n* = 41), but after RT, new fractures or increasing vertebral body sintering within the irradiated region occurred rarely (8%; *n* = 8). A pathological fracture before RT was negatively associated with stabilization 6 months after RT (OR 0.1, 95% CI 0.02–0.49, *p* = 0.004), while a Karnofsky performance score (KPS) ≥ 70% was associated positively with a stabilization effect through irradiation (OR 6.09, 95% CI 1.68–22.05, *p* = 0.006). Mean OS following first palliative RT was 10.7 months, and the KPS (≥70% vs. <70%) was shown to be a strong predictive factor for OS after RT (HR 0.197, 95% CI 0.11–0.35, *p* < 0.001). There was no significant difference in OS between patients with SCC and non-SCC. Palliative RT in symptomatic SBM of HNC provides sufficient symptom relief in the majority of patients, while only about one third of initially unstable SBM show re-stabilization after RT. Since patients in our multi-center cohort exhibited very limited OS, fractionation schemes should be determined depending on the patients’ performance status.

## 1. Introduction

Head and neck cancers (HNC) rank among the ten most common cancers worldwide with approximately 550,000 new patients diagnosed annually [1,2]. The majority of patients with HNC initially present with locoregionally confined disease that is usually treated with multimodal therapeutic approaches including surgery, radiotherapy (RT) and systemic treatments including chemotherapy or targeted agents [3,4,5,6].

The reported incidence of distant metastases in HNC, about 3% to 11%, is relatively low compared to other malignant tumors [7,8,9,10,11,12]. Several studies have reported that advanced T stage, lymph node metastases, extracapsular lymph node extension, locoregional recurrence, higher histologic grading, and hypopharyngeal/laryngeal locations are important risk factors for the development of distant metastases [7,8,9,10,11,12,13,14,15].

The lungs are by far the most common site for HNC metastasis [8,16], but other sites include the liver and bones [7,16]. In particular, bone metastases constitute a relatively rare finding in patients with metastatic HNC as compared to other malignancies. They can be found in up to 7% of patients, most frequently as vertebral disease and largely associated with metastases in other organs [7,17].

Owing to substantial advances in treatment over the last years, locoregional control and survival rates have improved significantly [3,4,5,18,19,20]. As a result, there may be increasing risks of developing bone metastases following the primary treatment. Associated complications, such as severe drug-resistant pain, pathological fractures, and/or neurologic deficits, can result in severe impairments of quality of life [21]. Moreover, the occurrence of bone metastases in HNC is often associated with a poor prognosis that needs to be considered in choosing therapeutic options [8].

In general, the treatment of spinal bone metastases (SBM) is complex and relies on multidisciplinary discussion of RT, surgery, systemic, symptomatic, and anti-resorptive therapies to reduce bone turnover. However, there is very limited evidence regarding actual treatment of SBM in HNC patients [17]. Based on bone metastasis data from other tumor sites, RT seems to be a key option, often in combination with systemic management [22,23].

It is well known that palliative RT achieves a considerable pain reduction in the majority of patients with symptomatic SBM [22]. Moreover, classifying the stability of SBM is a common clinical issue in the routine daily practice. In the case of unstable SBM, corset-based transient stabilization is often an initial measure taken to prevent pathological fractures. However, prescribed corsets lead to significant immobilization and restrictions in patients’ daily activities [24]. In addition, spondylosis and/or palliative RT and bisphosphonates are often performed to stabilize unstable SBM and to prevent skeletal-related complications. In recent years, our study group has demonstrated a significant effect of RT on stabilization of initially unstable spinal bone lesions for several histologies [25,26,27]. To date, only a few small studies and case reports have measured radiotherapeutic effects on SBM of HNC patients in terms of pain control and improvements of neurologic deficits from spinal cord compression, and to the best of our knowledge, there are no existing data concerning the impact of RT on the stability of SBM from metastatic HNC [17,28,29]. Therefore, this retrospective study aimed to assess pre- and post-RT stability of affected vertebral bodies, fracture rates, and survival, and to establish potential predictive factors for stability and survival in patients with metastatic HNC.

## 2. Material and methods

### 2.1. Patient Selection

The medical records of 66 patients with HNC-related SBM treated with palliative RT at the university hospitals of Heidelberg, Freiburg and Mainz between 2001 and 2019 were retrospectively analyzed. Patient data were collected from the cancer registries of participating centers. The diagnosis of SBM was based on radiological imaging (e.g., computed tomography [CT], magnetic resonance imaging [MRI], or bone scintigraphy). This analysis has been approved by the independent ethics committees of the medical faculties of the universities of Mainz, Freiburg, and Heidelberg.

### 2.2. Response Assessment

The stability of affected vertebral bodies was evaluated with the validated spinal instability neoplastic score (SINS) based on CT scans. Pre-RT treatment planning CTs and post-RT CT examinations at 3 and 6 months, that were regularly performed for metastasized HNC patients, were evaluated by a board-certified radiologist [30]. Several studies demonstrated that SINS constitutes a highly reliable, reproducible, and valid assessment tool to classify bone metastases in vertebral bodies as “stable“, “potentially unstable“, or “unstable“ [30,31]. The total score is calculated from six clinical and radiologic parameters comprising metastatic location, pain, bone quality, spinal alignment, vertebral body collapse, and posterolateral involvement of the metastatic vertebrae. In patients with more than one metastasis per irradiated region, the most critical bone metastasis was analyzed according to the SINS system. If several spinal regions were irradiated in a given patient, each region was evaluated separately in our analysis. During the three-monthly follow-up consultations, pain response, that was defined as a pain intensity decrease by ≥2 points on the numerical rating scale (NRS) or termination of analgesic medication was regularly assessed.

### 2.3. Treatment

RT was planned following CT simulation and performed by means of dorsal photon fields (6 or 18 MV photon energy). The planning target volume (PTV) included the metastatically affected vertebral body or bodies and the adjacent intervertebral discs. In many cases, it also included the caudally and cranially adjacent vertebral bodies. 

The median delivered dose was 30 Gy (range 20–42 Gy) in single fractions of 3 Gy (range 2–4 Gy). Many patients received additional systemic treatments such as chemotherapy, epidermal growth factor receptor (EGFR) antibodies, immunotherapy or antiresorptive therapies before, during, and after RT. In few patients, surgical interventions due to spinal cord compression or spinal instabilities were performed before or after RT.

### 2.4. Statistical Analysis

Statistical analysis was done using R software, version 4.0.0 (R Core Team 2020, Vienna, Austria). *p*-values of *p* < 0.05 were considered statistically significant. Overall survival (OS) was defined as the period from start of RT until death from any cause or until last follow-up. Survival after first palliative RT was analyzed with the Kaplan–Meier method and log-rank tests. Separate univariate Cox models for time-varying covariates and associated Wald tests were carried out to evaluate possible predictors of OS after RT. Furthermore, the association between vertebral body stability and multiple potential predictors was evaluated using logistic regression based on generalized estimating equations. The association between stabilization rates at 6 months after RT and pain response was displayed using cross tables, and chi-square tests were used for statistical analysis.

## 3. Results

A total of 66 patients with 95 target volumes and a total of 178 SBM (range 1–11 metastases per patient) of HNC that were treated with a total of 81 courses of RT were assessed according to SINS based on CT images before, as well as 3 and 6 months after irradiation. The mean follow-up time after first palliative RT was 8.1 months (range 0.3–85.0 months).

In total, 17 patients (26% of all patients) had a primary tumor originating from the salivary glands. The remaining primary tumors were mainly located in the pharynx, larynx, and oral cavity. Histologically, squamous cell carcinomas (SCC) dominated among the non-salivary gland tumors (46/49 patients), while adenoid-cystic carcinomas (ACC) and adenocarcinomas (AC) were most common among the salivary gland tumors (16/17 patients). Patient and treatment characteristics are summarized in Table 1; Table 2.

Besides SBM, pulmonary metastases comprised by far the most frequent metastatic localization (see Table 1), and distant metastases as well as skeletal metastases occurred metachronically in the majority of patients (75% and 79%, respectively).

Before the start of irradiation, SBM were associated with pain in 76% of irradiated lesions (72/95) and with a neurological deficit (i.e., metastatic neuropathy or spinal cord compression) in 22% (21/95, see Table 2). Pain response was documented in 63% of the treated lesions on medical records (43/68 SBM); however, in 4 patients with initially symptomatic SBM, there was no information on pain response to RT in follow-up. SBM-associated neurological deficits responded to RT in 47% of cases (9/19), taking into account 2 affected patients with missing clinical information in the follow-up (see Table 3).

The SINS as a validated and reliable method to quantify the stability of SBM takes the metastatic location, pain intensity, bone quality (osteolytic or osteoblastic), spinal alignment, vertebral body collapse, and posterolateral involvement into account. A total of 58 SBM (61% of all treated lesions) were rated as potentially unstable prior to RT according to the SINS, and only seven SBM (7%) were rated as unstable (i.e., SINS 13–18). Before the start of irradiation, the average SINS was 8.4 (standard deviation [SD] 2.6, range 4–15). As a result of the RT, the average SINS decreased to 6.3 (SD 2.8, range 2–14) after 3 months (*p* < 0.001, Wilcoxon–Mann–Whitney Test) and to 5.6 (SD 2.7, range 2–11) after 6 months (*p* < 0.001, Wilcoxon–Mann–Whitney Test). An improvement in the stability class according to SINS (i.e., from unstable to potentially unstable or from potentially unstable to stable) through RT was achieved in 12% and 16% of all treated lesions (11/95 and 15/95) within 3 and 6 months, respectively. Among patients with potentially unstable or unstable SBM still alive at 3 and 6 months after RT, a shift to a better stability class according to SINS was observed in 20% and 33% of the irradiated SBM respectively (11/55 and 10/30) (see Table 4 and Figure 1). The decrease of the average SINS mainly based on pain relief and mineralization effects, turning an osteolytic metastasis into a mixed or osteoblastic lesion. Notably, there was a significant correlation between stabilization rates after 6 months and pain relief: patients with a shift to an improved stability class according to SINS did more frequently benefit from pain response (*p* = 0.006, chi-square test).

However, two patients (3%) exhibited increased instability during the course of follow-up. In the first patient, SBM was initially assessed as stable and 3 months after RT as potentially unstable owing to a recent vertebral body collapse. In the second patient, SBM was initially classified as potentially unstable and acute paraplegia occurred during irradiation, requiring interruption of RT and decompression of the spinal cord by laminectomy.

Before initiation of RT, 41 SBM within target volumes exhibited pathological fractures; as such, post-RT fractures or increasing sintering of previously fractured vertebrae were diagnosed in only 8 SBM. However, when only patients with available follow-up imaging were considered, the rate of new fractures was 9% (5/56 SBM) and the rate of increasing vertebral body sintering within the irradiated region was 5% (3/56 SBM). In 63% of these post-RT fractures (5/8), SBM were initially assessed as potentially unstable and the remaining SBM as initially stable. In more than half of cases, patients with post-RT fractures (5/8) still exhibited a pain response after RT.

Overall, severe adverse events occurred in seven patients (11%) after RT, including six patients in whom eight irradiated SBM demonstrated a pathologic fracture and two patients in whom a new metastatic spinal cord compression occurred with consecutive cross-sectional symptoms requiring surgical intervention. Our analysis revealed that a pathological fracture before RT and a corset supply of SBM was negatively associated with the achievement of a stabilization effect through irradiation; furthermore, there was a significant positive association of immunotherapy and cetuximab administration with initial vertebral body stability and of the KPS ≥ 70% with the achievement of a stabilization effect 6 months after RT (for statistical details see Table 5).

In our cohort, poor survival was evident after palliative RT. By the end of follow-up, 82% of patients (54/66) had died. The mean OS after first palliative RT was 10.7 months (range 0.3–85 months), and the survival rates at 3, 6, and 12 months after RT amounted to 64%, 46%, and 33%, respectively. The Karnofsky Performance Score (KPS) was a strong predictive factor for OS after RT (HR 0.197, 95% CI 0.11–0.35, *p* < 0.001) (see Table 6).

The mean OS after first RT of patients with a KPS < 70% and KPS of ≥ 70% were 4.3 months (95% CI 1.5–4.5 months) and 15.8 months (95% CI 9.0–23.4 months), respectively (see Figure 2). 

There was no significant difference in OS between patients with SCC and non-SCC (see Figure 3).

Furthermore, the cumulative radiation dose administered showed a positive association with OS after RT (HR 0.95, 95% CI 0.92–0.99, *p* = 0.016).

In contrast, the prevalence of additional liver metastases, lung metastases, brain metastases, visceral metastases, non-visceral metastases, fractures and the use of chemotherapy, immunotherapy, cetuximab, or bisphosphonates were not statistically significant for prediction of OS after RT (see Table 6).

## 4. Discussion

In our analysis, SBM-related pain symptoms and neurologic deficits responded to RT in 63% and 47%, respectively. The evaluation of the SINS showed that palliative RT reduced the score on average by about 2 points after 3 months and by about 3 points after 6 months compared to the initial value before RT. Accordingly, a shift to a better stability class according to SINS was observed in 20% and 33% of initially unstable or potentially unstable SBM among surviving patients at 3 or 6 months after palliative RT, respectively. Only a minority of patients (3%) exhibited increased instability during the course of follow-up. While pathological fractures of SBM were commonly detected before the initiation of RT, treatment did only rarely result in increasing fracture rates and/or new metastatic spinal cord compression with consecutive cross-sectional symptoms. In sum, a total of seven patients (11%) developed serious adverse advents after RT, which is in line with the results of other study groups [32,33].

In previous studies from our working group, we investigated recalcification rates of osteolytic SBM after palliative RT in several other tumor histologies and found substantially differing re-calcification rates. For instance, the majority of unstable SBM from metastatic breast cancer and other gynecologic malignancies stabilized by 6 months post-RT [26,27]. Conversely, substantial stabilization of SBM from pulmonary cancers occurred in only one quarter of patients [25], and corresponding rates from malignant melanoma, renal cell cancer and colorectal cancer were considerably worse [34,35,36]. The differing re-ossification rates may partly be explained by individual radiation sensitivities of the respective tumor types. However, the exact mechanism for recalcification of bone lesions following RT is only incompletely understood, and further co-factors, such as synergistic effects of systemic treatments, tumor micromilieu, different scoring systems, and additional patient characteristics, may contribute to the observed different recalcification effects [37]. In our analysis, pathologic fractures of SBM prior to RT and a corset prescription were found to correspond to inferior stabilization rates; in addition, the administration of immunotherapy and cetuximab before starting RT was associated with higher numbers of SBM scored stable pre-RT than without these treatment approaches. Since patients with unstable SBM primarily receive a corset fitting, it can be assumed that the statistically significant negative effect on the stabilization rates of initially unstable SBM is primarily an indication effect.

In our study, only a limited number of patients derived a benefit from palliative RT with regard to stabilization, mainly related to the limited life expectancy with distant metastases. Specifically in this study, only 46% of patients were still alive at 6 months following RT. Mean OS amounted to only 10.7 months after first palliative RT [38,39]. This is readily explained by the fact that HNC bone metastases usually represent end-stage disease, with the majority of patients having further extraskeletal metastases at the time of osseous disease diagnosis [8]. In our cohort, 70% of patients presented with additional visceral metastases especially in the lungs and liver. However, the presence of extraskeletal metastases did not statistically influence OS after RT in our analysis.

It is crucial to identify factors that serve as prognosticators for increased survival, as only patients with a life expectancy of at least 3–6 months may experience the bone-mineralizing and stabilizing effects of palliative RT. Consistent with the findings of several other studies [28,36,40,41,42], we identified the performance status as a strong prognostic factor for predicting OS after RT. As recalcification of irradiated osteolytic SBM usually takes up to several months, it is unlikely that patients with KPS < 70% would have a benefit with regard to stabilization within the remaining lifespan. In our study, 33 patients presented initially with a KPS < 70% and unstable SBM, and only a minority of them (*n* = 3) experienced a stabilization effect after palliative RT. In contrast, patients with a good performance status exhibited a higher probability for significant re-ossification and stabilization due to a longer life expectancy and continued mobility-related physical strain to the bones. Vice versa, unstable and painful SBM as well as SBM causing neurologic symptoms may directly have an impact on the patients’ KPS, as these factors may deteriorate the ability to perform activities of daily living. Therefore, KPS is both prognostic for prolonged survival, thereby increasing the probability for re-ossification and stabilization after RT, and correlating with the SINS, as pain and immobilization directly affect the KPS. In our cohort, 17 patients presented with a KPS ≥ 70% and unstable SBM, and more than half of them (*n* = 9) stabilized within 6 months.

Interestingly, bisphosphonate administration did not lead to increased stabilization rates after palliative RT in our cohort. In contrast, Grisanti and colleagues [33] showed that the combination of RT and bone-modifying treatment (bisphosphonates and denosumab) resulted in significantly improved survival in nasopharyngeal carcinoma patients, when compared to each treatment alone. Furthermore, the authors could not detect severe adverse reactions following bone-modifying treatment and concluded that the combination of bone-modifying treatment and RT may be an appropriate approach for bone metastases of nasopharyngeal carcinoma patients. However, it should be noted that the biology and the prognosis of nasopharyngeal carcinoma differs from other HNCs: Therefore, concerning the results of this large study and our findings, further investigations are necessary to clarify which patients with HNC-related SBM benefit from bone-modifying treatment initiation after palliative RT.

Moreover, the applied cumulative radiation dose was associated with an improved OS after palliative RT of SBM, which is supported by another study [43]. However, since the selection of radiation dose and fractionation highly depends on the general condition of the patient, the radiation dose cannot be evaluated with certainty as an independent predictive factor.

While being one of the largest multi-center analyses for HNC bone metastases, our study nevertheless has some limitations, among them the retrospective assessment of the dataset. Secondly, HNCs in our study comprise diverse histologies, including SCC, AC, ACC, sarcomas, neuroblastoma, and transitional cell carcinoma; these vary widely in aggressiveness and radiation sensitivity. However, our analysis showed no significant difference between patients with SCC and non-SCC in terms of survival after RT. Third, owing to the limited number of patients, a statistically sound multivariate analysis of prognostic factors for OS after RT was not possible. Fourth, it was not feasible to retrospectively assess quality of life in our cohort, which is an important parameter, especially when considering the palliative treatment intention. Prospective trials, focusing on the quality of life for RT of HNC-related SBM, ideally within a multicentric approach, are therefore warranted in future.

In summary, we could show that only a minority of patients with unstable SBM and KPS < 70% had a benefit in terms of stabilization after palliative RT due to very poor life expectancy; thus, single fraction RT or strongly hypofractionation regimens appear preferable for this subgroup of patients, since they provide pain control comparable to protracted radiation schedules and avoid unnecessary hospitalization [22]. Especially in the context of the palliative intention of RT, patients may appreciate the reduced treatment time for shortened fractionation regimens. In contrast, our results favor the use of protracted radiation schedules for patients with unstable SBM and KPS of ≥70%, since about half of the affected patients experienced significant re-ossification and a stabilization effect during 6-month follow-up after palliative RT.

## 5. Conclusions

Palliative RT for symptomatic SBM of HNC patients provides sufficient symptom relief in the majority of patients, while residual life expectancy is very poor. As a consequence, the chance to achieve significant stabilization following palliative irradiation of unstable spinal bone lesions is limited, particularly for patients with a KPS less than 70%. Therefore, the fractionation schedule should be primarily based on the performance status, with single fraction RT advisable for patients with KPS < 70% to avoid unnecessary hospitalization in the remaining short survival time.

## Figures and Tables

**Figure 1 cancers-12-01950-f001:**
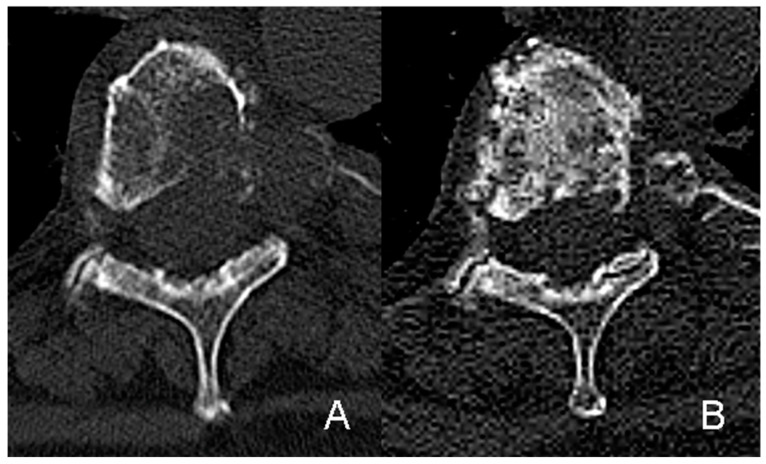
Extensive osteolytic metastasis in the 5th thoracic vertebra including the dorsal part of the corresponding left costa of an adenoid-cystic carcinoma arising from the soft palate, rated as potentially instable according to SINS (**A**). Significant re-ossification and stabilization of the osteolytic bone lesion 3 months after radiation treatment (**B**).

**Figure 2 cancers-12-01950-f002:**
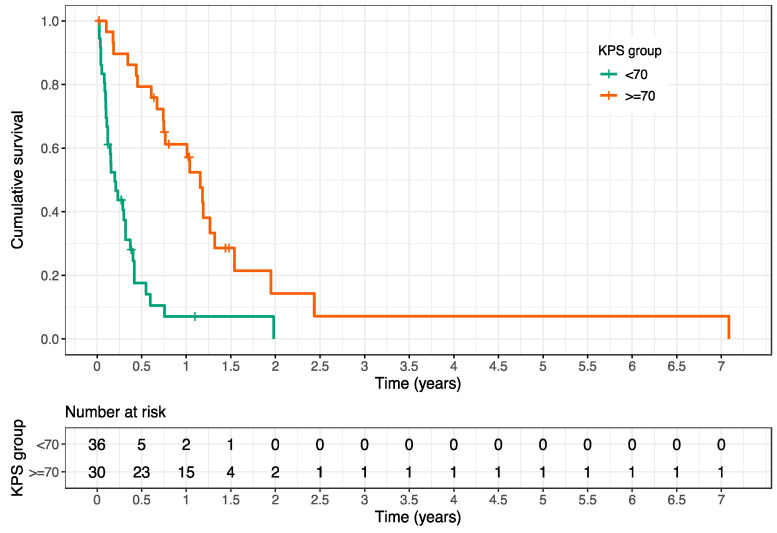
Kaplan–Meier estimation of overall survival after first palliative radiotherapy depending on the performance score showing a significantly improved prognosis for patients with a Karnofsky Performance Score (KPS) of ≥ 70% compared to patients with a KPS < 70% (*p* < 0.001, log-rank test).

**Figure 3 cancers-12-01950-f003:**
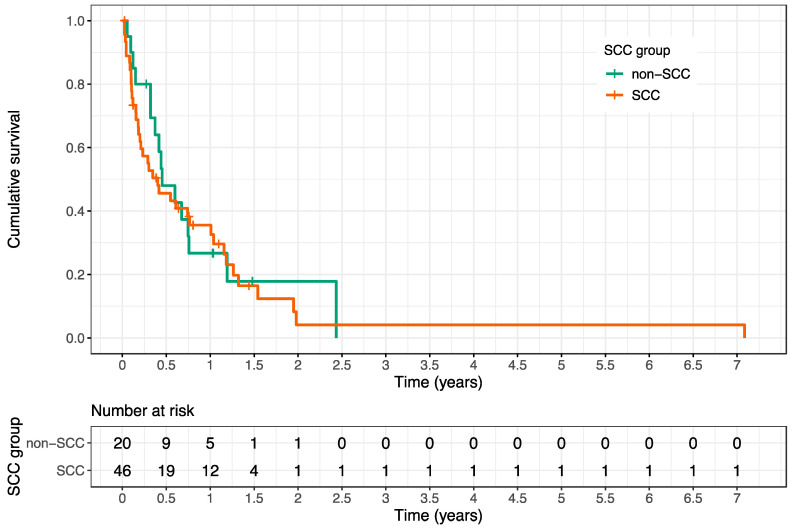
Kaplan–Meier estimation of overall survival of squamous cell carcinoma (SCC) patients compared to non-SCC patients after first palliative radiotherapy (RT). The 3, 6, and 12 month survival rates for SCC patients after first palliative RT were 57% (95% CI 44–74%), 46% (95% CI 33–63%) and 36% (95% CI 24–53%), while the corresponding survival rates for non-SCC patients were 80% (95% CI 64–100%), 48% (95% CI 30–77%) and 27% (95% CI 13–56%), respectively (*p* = 0.7, log-rank test).

**Table 1 cancers-12-01950-t001:** Patients’ characteristics.

Characteristics	Value	%
**Age at start of RT (years)**		
Median (range) 60.8 (22.2–80.0)		
**Gender**		
Female	13	19.7
Male	53	80.3
**Karnofsky PS**		
<70%	47	58.0
≥70%	34	42.0
**Number of bone metastases**		
Median (range) 2.5 (1–11)		
Solitary	36	37.9
Multiple	59	62.1
**Spine involvement**		
Cervical	9	9.5
Cervicothoracic	8	8.4
Thoracic	38	40.0
Thoracolumbar	17	17.9
Lumbar	15	15.8
Lumbosacral	2	2.1
Sacral	6	6.3
**Histology**		
Adenocarcinoma	3	4.5
Adenoid cystic carcinoma	13	19.7
Squamous cell carcinoma	46	69.7
Sarcoma	2	3.0
Transitional cell carcinoma	1	1.5
Neuroblastoma	1	1.5
**Distant extraskeletal metastases**		
Brain	6	7.4
Lung	53	65.4
Liver	17	30.9
Adrenal gland	2	2.5
Visceral	57	70.4
Non-visceral	24	29.6

Abbreviations: RT = Radiotherapy, Karnofsky PS = Karnofsky performance score, SBM = Spinal bone metastases.

**Table 2 cancers-12-01950-t002:** Treatment.

Characteristics	Value	%
**RT completed**		
Yes	86	90.5
No	9	9.5
**Radiation dose**		
**RT completed**		
Cumulative dose		
Median	30	
Range	8–60	
**RT discontinued**		
Cumulative dose		
Median	15	
Range	3–27	
**Indications for palliative RT**		
Pain	72	75.8
Instability	64	67.4
Neurological impairment	21	22.1
Postoperative	13	13.7
**Other treatments**		
Orthopedic corset	25	26.3
Bisphosphonates	35	43.2
Chemotherapy	55	67.9
Prior to RT	42	51.9
After RT	46	56.8
Cetuximab	34	42.0
Prior to RT	27	33.3
After RT	25	30.9
Immunotherapy	13	16.0
Prior to RT	5	6.2
After RT	12	14.8
Surgery	16	16.8
Prior to RT	13	13.7
Laminectomy	6	6.3
Spondylodesis	5	5.3
Both	2	2.1
After RT	3	3.2
Laminectomy	1	1.1
Spondylodesis	2	2.1

Abbreviations: RT = radiotherapy, Gy = Gray.

**Table 3 cancers-12-01950-t003:** Assessment of neurological impairments.

Parameter	*n*	%
Frankel classification before RT		
No deficit (E)	74	77.9
Minor motor or sensory deficit (D)	18	18.9
Major motor or sensory deficit (A, B, C)	3	3.2
Frankel classification after RT		
No deficit (E)	83	87.4
Minor motor or sensory deficit (D)	7	7.4
Major motor or sensory deficit (A, B, C)	3	3.2
NA	2	2.1

Abbreviations: RT = radiotherapy, NA = not analyzable.

**Table 4 cancers-12-01950-t004:** Results of the stability assessment according to SINS.

Parameter	*n*	%
Stability before RT		
Unstable	64	67.4
Stable	31	32.6
Stability after 3 months		
Unstable	23	24.2
Stable	32	33.7
NA	40	42.1
Stability after 6 months		
Unstable	10	10.5
Stable	20	21.1
NA	65	68.4

Abbreviations: RT = radiotherapy, NA = Not analyzable, because the follow-up examination was missing due to a deterioration of the general condition or death.

**Table 5 cancers-12-01950-t005:** Univariate Cox regression analysis of prognostic factors related to stability of SBM prior to and after RT.

Predictor	*p*-Value	OR	CL
Karnofsky PS ≥ 70%			
- Stable SBM pre-RT	<0.001	4.16	1.81–9.57
- Stable SBM 3 mo. post-RT	0.29	1.84	0.60–5.64
- Stable SBM 6 mo. post-RT	0.006	6.09	1.68–22.05
Chemotherapy (yes)			
- Stable SBM pre-RT	0.088	2.42	0.88–6.71
- Stable SBM 3 mo. post-RT	0.099	3.53	0.79–15.75
- Stable SBM 6 mo. post-RT	0.75	0.73	0.10–5.13
Immunotherapy (yes)			
- Stable SBM pre-RT	0.002	4.92	1.82–13.28
- Stable SBM 3 mo. post-RT	0.74	2.86	0.55–14.90
- Stable SBM 6 mo. post-RT	0.61	v	0.26–9.57
Cetuximab (yes)			
- Stable SBM pre-RT	0.008	3.25	1.37–7.72
- Stable SBM 3 mo. post-RT	0.36	1.23	0.36–4.17
- Stable SBM 6 mo. post-RT	0.56	0.64	0.14–2.86
Bone-modifying therapy (yes)			
- Stable SBM pre-RT	0.23	1.74	0.70–4.32
- Stable SBM 3 mo. post-RT	0.58	0.73	0.24–2.22
- Stable SBM 6 mo. post-RT	0.38	0.46	0.09–2.54
Total dose of palliative RT			
- Stable SBM pre-RT	0.05	1.06	1.00–1.12
- Stable SBM 3 mo. post-RT	0.10	1.06	0.99–1.15
- Stable SBM 6 mo. post-RT	0.16	1.07	0.97–1.18
Liver metastases (yes)			
- Stable SBM pre-RT	0.35	0.61	0.21–1.71
- Stable SBM 3 mo. post-RT	0.64	1.35	0.39–4.69
- Stable SBM 6 mo. post-RT	0.18	3.92	0.52–29.59
Lung metastases (yes)			
- Stable SBM pre-RT	0.58	1.30	0.52–3.26
- Stable SBM 3 mo. post-RT	0.79	1.17	0.37–3.73
- Stable SBM 6 mo. post-RT	0.73	1.30	0.29–5.85
Brain metastases (yes)			
- Stable SBM pre-RT	0.64	0.72	0.18–2.90
- Stable SBM 3 mo. post-RT	NA	NA	NA
- Stable SBM 6 mo. post-RT	NA	NA	NA
Visceral metastases (yes)			
- Stable SBM pre-RT	0.68	1.23	0.47–3.17
- Stable SBM 3 mo. post-RT	0.59	1.38	0.43–4.46
- Stable SBM 6 mo. post-RT	0.48	1.74	0.38–8.09
Non visceral metastases (yes)			
- Stable SBM pre-RT	0.95	1.03	0.40–2.64
- Stable SBM 3 mo. post-RT	0.44	1.71	0.44–6.62

Abbreviations: Karnofsky PS = Karnofsky performance score, OR = odds ratio, CL = confidence limits of the results for a confidence level of 95%, RT = radiotherapy, SCC = squamous cell carcinoma, SBM = spinal bone metastases; mo. = months.

**Table 6 cancers-12-01950-t006:** Univariate Cox regression analysis of prognostic factors related to overall survival after RT.

Parameter	*p*-Value	HR	CL
Karnofsky PS	<0.001	0.197	0.11–0.35
(≥70% vs. <70%)			
Chemotherapy	0.45	0.79	0.44–1.45
(yes vs. no)			
Immunotherapy	0.13	0.45	0.16–1.27
(yes vs. no)			
Cetuximab	0.46	1.24	0.45–1.44
(yes vs. no)			
Bone-modifying therapy	0.4	0.79	0.45–1.37
(yes vs. no)			
Total dose	0.016	0.95	0.92–0.99
(palliative RT)			
Liver metastases	0.075	1.66	0.95–2.88
(yes vs. no)			
Lung metastases	0.21	1.44	0.81–2.55
(yes vs. no)			
Brain metastases	0.45	1.4	0.59–3.30
(yes vs. no)			
Visceral metastases	0.098	1.67	0.91–3.07
(yes vs. no)			
Non visceral metastases	0.79	1.09	0.57–2.11
(yes vs. no)			
Pathologic fracture	0.71	0.9	0.65–1.90
(yes vs. no)			
Histology SCC	0.67	1.13	0.65–1.98
(yes vs. no)			
Salivary gland tumors	0.35	0.76	0.43–1.35
(yes vs. no)			

Abbreviations: Karnofsky PS = Karnofsky performance score, HR = hazard Ratio, CL = confidence limits of the results for a confidence level of 95%, RT = radiotherapy, SCC = squamous cell carcinoma.

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
