# Peer review of "The Role of Palliative Radiotherapy in the Treatment of Spinal Bone Metastases from Head and Neck Tumors—A Multicenter Analysis of a Rare Event"

_cancers, 2020, doi:10.3390/cancers12071950_

Round 1
Reviewer 1 Report
Manuscript Cancers-853606
Reviewer comments
The manuscript entitled "The role of palliative radiotherapy in the treatment of spinal bone metastases from head and neck tumors – A multicenter analysis of a rare event" by Tilman Bostel et al., presents original data on the multicenter retrospective analysis of the impact of radiotherapy on overall survival and stability in patients with spinal bone metastases from head and neck tumors in order to identify predictive factors for stability and survival of patients.
Presented results are very interesting with an important cohort when considering the very low incidence of bone metastases from head and neck tumors. The reviewer has however several minor comments that need to be addressed.
Minor comments
Abstract line 40: “…Individual RTs… What does it mean if prior to RT? Patients? Not clear! Sentence needs rephrasing.
Abstract line 43: same question. Individual RTs means patients?
35 patients have received bisphosphonates treatments to stabilize the bone but according to table 5 such treatments have no consequence on the benefit of RT induced stability. Does it mean that RT effect is additive to bisphosphonate effect (so independent) or that bisphosphonate has no stabilizing effect on SBM? It may be interesting to discuss this point with the recalcification line 275!
The Reviewer will appreciate the comments to be taken into account in a revised version of the manuscript.
Author Response
Reviewer #1:
The manuscript entitled "The role of palliative radiotherapy in the treatment of spinal bone metastases from head and neck tumors – A multicenter analysis of a rare event" by Tilman Bostel et al., presents original data on the multicenter retrospective analysis of the impact of radiotherapy on overall survival and stability in patients with spinal bone metastases from head and neck tumors in order to identify predictive factors for stability and survival of patients.
Presented results are very interesting with an important cohort when considering the very low incidence of bone metastases from head and neck tumors. The reviewer has however several minor comments that need to be addressed.
- We thank the reviewer for reviewing our paper and for the favorable feedback.
Minor comments
Abstract line 40: “…Individual RTs… What does it mean if prior to RT? Patients? Not clear! Sentence needs rephrasing.
- Since 66 patients with 95 target volumes comprising a total of 178 individual spinal bone metastases were included in our analysis, we tried to point out that the prevalence of pain and neurologic deficits were measured according to the number of irradiated target volumes. In order to clarify, we followed the suggestion of the reviewer and have rephrased into irradiated lesions or treated lesions.
Abstract line 43: same question. Individual RTs means patients?
- As explained above, we analyzed pain symptoms and neurologic deficits for each spinal bone metastasis. Patients often suffered from several spinal bone metastases, and pain severity and neurology symptoms were assessed for each irradiated target volume.
35 patients have received bisphosphonates treatments to stabilize the bone but according to table 5 such treatments have no consequence on the benefit of RT induced stability. Does it mean that RT effect is additive to bisphosphonate effect (so independent) or that bisphosphonate has no stabilizing effect on SBM? It may be interesting to discuss this point with the recalcification line 275!
- The reviewer raises an important topic. Based on our data, there was no additional beneficial effect of bisphosphonate treatment in terms of radiotherapy-induced stabilization. However, we did not analyze the effects of bisphosphonates on unirradiated bone metastases, where bisphosphonates may have had an effect. It should be noted that bisphosphonates were administered in about one third of cases (n=35, 43.2%) and that only 30 lesions could be assessed at 6 months after palliative radiotherapy (considering the rather poor patient survival), making meaningful statistical analyses more difficult, so that the results should be interpreted with caution. There are conflicting results presented by Grisanti et al. who showed that the combination of RT and bone-modifying treatment (bisphosphonates and denosumab) led to significantly improved survival in nasopharyngeal carcinoma patients, when compared to each treatment alone. As they also could not detect higher-graded toxicities related to bone-modifying treatment, the authors concluded that the combination of bone-modifying treatment and RT may be an appropriate approach for bone metastases of nasopharyngeal carcinoma patients. Concerning the conflicting data, further investigations are necessary to clarify which patients with HNC-related SBM benefit from bone-modifying treatment initiation after palliative RT. We thank the reviewer for pointing out this issue, and we therefore have pointed out this interesting point and its implications/limitations in the revised manuscript.
Reviewer 2 Report
The authors present a multicenter retrospective regarding the effect of palliative radiotherapy on the stability of spinal metastases from head and neck cancers (HNC). The authors should be applauded for collecting data on 66 patients with metastases from HNC. The authors correctly point out that this group often not included in larger studies. I have the following comments/concerns regarding the study:
Methods:
- Unfortunately because of the rarity of the disease the patients were included over a very large time span.
- was it standard of care for patients to have 3 and 6 months post RT to have a followup CT-scan? please clarify.
- myelon should be spinal cord
- Please clarify how response to RT was assessed
Results/discussion:
- please state clearly in the results if the number of target volumes were used for the analyses or the number of patients
- If in 76% of the patients pain was present, what was the indication for treatment in the other 24%?
- The authors state that a pain response was noted in 63% of the patients, what does this mean? full response? 0.1 down in VAS score? medication change?
- It would be great if the authors could demonstrate a break down of the SINS components and not only the different categories of stability.
- It would be great if the authors could comment on the factors that resulted in the decrease of the SINS score? was it because of lytic turning into blastic lesion etc
- Most importantly, although the patients imaging showed a shift in stability category did anything change for the patients in terms of pain, QOL etc. I realize that this the caveat of this study, being a retrospective study makes it hard to incorporate QOL scores yet this is really critical for the implications of the results. The authors should comment on this further in their discussion or acknowledge the absence of the QOL measures.
- Similar, the authors mention fractures post RT yet are these clinically relevant? Did anything change for the patient?
- It would also be interesting to know if the change in stability assessment was associated with the pain response.
- The authors state the association between KPS and stability yet would it not be as a result of a stable SBM the patient is able to maintain their KPS as there is no mechanical problem?
- As mentioned above the authors should highlight the clinical implications better in their discussion.
Overall really interesting study yet with known limitations due to the retrospective study design.
Author Response
Reviewer #2:
The authors present a multicenter retrospective regarding the effect of palliative radiotherapy on the stability of spinal metastases from head and neck cancers (HNC). The authors should be applauded for collecting data on 66 patients with metastases from HNC. The authors correctly point out that this group often not included in larger studies.
- We would like to thank the reviewer for the efforts to review our paper and for the valuable comments.
I have the following comments/concerns regarding the study: Methods:
Unfortunately because of the rarity of the disease the patients were included over a very large time span.
- We fully agree with the reviewer that the inclusion time span ranging between 2001 and 2019 is relatively long. However, the large inclusion time interval was chosen in order to obtain a sufficient number of patients for statistical analyses. The longer time span may especially be relevant, if there were considerable alterations in terms of radiotherapy or systemic treatment over the time, leading to increased survival rates. As immunotherapy was no first-line treatment until very recently, the vast majority of analyzed patients did not receive checkpoint inhibitors. The majority of patients received either single-agent chemotherapy or the EXTREME treatment regimen in case of good performance status, which has been relatively consistent in the last decade. Although the elongated time span may bias our results, we nevertheless feel that resulting number of patients included in our analysis helps to provide meaningful statistics for a very rare event.
Was it standard of care for patients to have 3 and 6 months post RT to have a followup CT-scan? please clarify.
- It was standard of care in all three centers to perform follow-up CT scans within the regular staging examinations, which were conducted three-monthly for metastasized head-and-neck cancer patients.
Myelon should be spinal cord
- We thank the reviewer for this comment and have changed the term in the revised version.
Please clarify how response to RT was assessed
- We apologize it this information was not clearly stated in the initial manuscript. Response to radiotherapy was evaluated in the follow-up consultations both clinically by assessing pain response (records in the patient files) and using cross-sectional imaging. In order to standardize the assessment of spinal bone metastases, all CT scans were read and evaluated by a board-certified radiologist. We have clarified this point in the revised version of the manuscript.
Results/discussion: Please state clearly in the results if the number of target volumes were used for the analyses or the number of patients
- For demographic analyses and survival measurements, the total number of patients (n=66) were used as reference for calculations. Analyses of pain response and stability rates were based on the number of irradiated lesions (n=95), as a significant number of patients received radiotherapy for multiple spinal bone metastases over the course of their disease. As requested, we have clarified when the total number of patients and when the number of treated lesions were used as reference in the Results section of our manuscript.
If in 76% of the patients pain was present, what was the indication for treatment in the other 24%?
- Severe spinal bone metastasis-related pain, neurological symptoms as well as vertebral instability (impending or manifest vertebral fractures) were indications for radiotherapy. Patients with no pain present at the time of radiotherapy initiation either suffered from neurological symptoms or exhibited unstable spinal bone metastases that harbored a significant risk for fractures and/or neurologic symptoms or had already fractured. In some cases, radiotherapy was administered in an adjuvant setting (e.g., after spondylodesis) for further stabilization. Table 2 shows the percentage of the different indications for radiotherapy in our dataset.
The authors state that a pain response was noted in 63% of the patients, what does this mean? full response? 0.1 down in VAS score? medication change?
- A clinically meaningful pain response was assumed if the numeric rating scale (NRS=10-point VAS) decreased by ≥2 points. In some cases, retrospective assessment of this effect was challenging to perform, wherefore also medication alterations were tried to survey, and termination of analgesic medication was considered as pain response, too. We have now clarified this point in the revised version.
It would be great if the authors could demonstrate a break down of the SINS components and not only the different categories of stability.
- We followed the reviewer’s suggestion and mentioned the different components of the SINS score both in the Material and Methods and in the Results section
It would be great if the authors could comment on the factors that resulted in the decrease of the SINS score? was it because of lytic turning into blastic lesion etc
- We thank the reviewer for this valid question. The decrease of the SINS score after radiotherapy treatment was mainly based on pain relief and/or stabilization effects (e.g., mineralization of an osteolytic lesion). Only in few cases, the SINS increased after radiotherapy, and this effect was mainly based on progressing vertebral body collapse or metastatic progression with de novo posterior spinal element involvement. We have also mentioned these aspects in the revised manuscript.
Most importantly, although the patients imaging showed a shift in stability category did anything change for the patients in terms of pain, QOL etc. I realize that this the caveat of this study, being a retrospective study makes it hard to incorporate QOL scores yet this is really critical for the implications of the results. The authors should comment on this further in their discussion or acknowledge the absence of the QOL measures.
- The reviewer is absolutely right regarding the importance of qualify of life (QoL) for this treatment indication. While we made several efforts to accurately assess pain dynamics in treated patients, assessment of QoL was not possible due to the retrospective nature of the study. Since nearly all patients had died by the time of data analysis, we were not able to perform retrospective QoL surveys by contacting the patients. As we agree with the reviewer in terms of the relevance of this point, we have added the missing QoL data as limitation of our study, clearly stating that prospective studies are warranted to investigate this topic in the context of head-and-neck cancer-related spinal bone metastases in the future.
Similar, the authors mention fractures post RT yet are these clinically relevant? Did anything change for the patient?
- In order to answer this question, we again carefully examined the medical records of patients who developed new fractures after radiotherapy. Interestingly, in more than half of cases (5 of 8), patients with post-RT fractures still exhibited a pain response after RT. We have added this information in the revised manuscript.
It would also be interesting to know if the change in stability assessment was associated with the pain response.
- We thank the reviewer for this valuable suggestion. We have analyzed the interaction between stabilization rate (according to the SINS) and pain response using crosstab statistics and chi-square tests. Here, we could demonstrate a significant correlation between stabilization and pain relief (p=0.006) in our dataset. It should be noted that the pain intensity is one of the components of the Spine Instability Neoplastic Score, which of course influences the interaction analyses. However, we have now presented the significant correlation between both parameters in our revised version of the manuscript.
The authors state the association between KPS and stability yet would it not be as a result of a stable SBM the patient is able to maintain their KPS as there is no mechanical problem?
- We agree with the reviewer that unstable spinal bone metastases may directly affect patients’ performance status, as such unstable spinal bone metastases can cause both immobility and pain, affecting the patients’ ability to perform certain activities of daily living without the help of others. On the other hand, patients with superior Karnofsky performance status have been shown to exhibit a longer life expectancy and could therefore rather benefit from the re-stabilizing effects of palliative radiotherapy. We now emphasized both aspects more clearly in the revision of our manuscript.
As mentioned above the authors should highlight the clinical implications better in their discussion.
- We followed the suggestion and revised the Discussion in order to clearly present the clinical implications that may be concluded based on our results. In brief, there are three main clinical implications:
1) Due to the poor life expectancy of patients with unstable spinal bone metastases and poor Karnofsky performance status, palliative radiotherapy should mainly focus on pain relief for these patients, wherefore single-fraction or strongly hypofractionated schedules (i.e., 1x8 Gy or 5x4 Gy) that provide equivalent pain responses are appropriate approaches to shorten the treatment time without abrogating the analgesic effect of radiotherapy.
2) It is essential to a priori identify patients who exhibit a longer survival rate and who therefore may benefit from fractionated radiotherapy (e.g., 10x3 Gy) regarding bone-stabilizing effects. Despite presenting one of the largest retrospective series regarding radiotherapy for head-and-neck cancer-related spinal bone metastases, we were not able to identify additional prognostic parameters for overall survival besides the performance status. Therefore, the present data underline the importance of this parameter in the treatment decision process, whether standard fractionation radiotherapy or shortened regimens should be used.
3) As bone-modifying agents (bisphosphonates and denosumab) had no statistically significant effect on the stabilization rate, their usage should be critically evaluated in patients with short survival rates, as these agents harbor their beneficial effects in terms of mineralization rather over the long term. Grisanti and colleagues could show a benefit from the combination of bone-modifying agents and radiotherapy for nasopharyngeal carcinoma bone metastases, although this disease’s biology and prognosis varies considerably from other head-and-neck cancers. Therefore, further studies are necessary to reveal the role of these agents in the context of head-and-neck cancer-related spinal bone metastases.
Overall really interesting study yet with known limitations due to the retrospective study design.
- We thank the reviewer again for this feedback. Although one should be aware of the limitations of retrospective analyses, retrospective studies are the only possibility to increase the evidence, if prospective trials are missing. We have presented the results of one of the largest studies consisting of head-and-neck tumor-derived spinal bone metastases treated by radiotherapy. However, we addressed this point and mentioned the limitations of our study in the Discussion.